# All-Atom Molecular Dynamics Simulations Indicated the Involvement of a Conserved Polar Signaling Channel in the Activation Mechanism of the Type I Cannabinoid Receptor

**DOI:** 10.3390/ijms24044232

**Published:** 2023-02-20

**Authors:** Arijit Sarkar, Argha Mitra, Attila Borics

**Affiliations:** 1Laboratory of Chemical Biology, Institute of Biochemistry, Biological Research Centre, 62 Temesvári krt., H-6726 Szeged, Hungary; 2Theoretical Medicine Doctoral School, Faculty of Medicine, University of Szeged, 97 Tisza L. krt., H-6722 Szeged, Hungary

**Keywords:** GPCR, cannabinoid, activation mechanism, signal transduction, signaling pathway, G protein, arrestin, molecular dynamics

## Abstract

The type I cannabinoid G protein-coupled receptor (CB1, GPCR) is an intensely investigated pharmacological target, owing to its involvement in numerous physiological functions as well as pathological processes such as cancers, neurodegenerative diseases, metabolic disorders and neuropathic pain. In order to develop modern medications that exert their effects through binding to the CB1 receptor, it is essential to understand the structural mechanism of activation of this protein. The pool of atomic resolution experimental structures of GPCRs has been expanding rapidly in the past decade, providing invaluable information about the function of these receptors. According to the current state of the art, the activity of GPCRs involves structurally distinct, dynamically interconverting functional states and the activation is controlled by a cascade of interconnecting conformational switches in the transmembrane domain. A current challenge is to uncover how different functional states are activated and what specific ligand properties are responsible for the selectivity towards those different functional states. Our recent studies of the μ-opioid and β_2_-adrenergic receptors (MOP and β_2_AR, respectively) revealed that the orthosteric binding pockets and the intracellular surfaces of these receptors are connected through a channel of highly conserved polar amino acids whose dynamic motions are in high correlation in the agonist- and G protein-bound active states. This and independent literature data led us to hypothesize that, in addition to consecutive conformational transitions, a shift of macroscopic polarization takes place in the transmembrane domain, which is furnished by the rearrangement of polar species through their concerted movements. Here, we examined the CB1 receptor signaling complexes utilizing microsecond scale, all-atom molecular dynamics (MD) simulations in order to see if our previous assumptions could be applied to the CB1 receptor too. Apart from the identification of the previously proposed general features of the activation mechanism, several specific properties of the CB1 have been indicated that could possibly be associated with the signaling profile of this receptor.

## 1. Introduction

The type I cannabinoid receptor (CB1) belongs to the family of class A G protein-coupled receptors (GPCRs), a remarkably large protein superfamily with high pharmacological significance [1,2,3]. CB1 receptors were first cloned from rat cerebral cortex, which then initiated intensive examinations of the endocannabinoid system [4]. As an outcome, the CB1 receptor was identified to participate in many vital physiological functions as well as pathological processes. Consequently, this receptor was introduced as a promising target for drugs and therapies against a variety of ailments, including different cancers, neurodegenerative diseases, metabolic disorders and neuropathic pain [5,6].

The extracellular activation of GPCRs (including the CB1 receptor) through the binding of an endogenous or exogenous ligand usually stimulates more than one intracellular signaling pathways. These pathways are mediated by either G proteins or arrestins, and one them is usually associated with some undesired side effects. A recent challenge of modern pharmaceutical research is, therefore, to facilitate the design of biased, or in other words pathway-specific, GPCR ligands. The therapeutic potential of biased agonists lies in their capacity to selectively activate the relevant pathway and, therefore, to suppress the development of harmful side effects. The development of such high-affinity, high-efficacy and functionally selective GPCR ligands necessitates in-depth understanding of the structural mechanism of receptor activation and the formulation of a quantitative structure–activity model.

The pool of atomic-resolution three-dimensional structures of GPCRs has been growing exponentially in the past decade owing to breakthrough developments in experimental techniques and procedures. These structures are now readily available in the Brookhaven Protein Data Bank and curated in the GPCRdb database (http://gpcrdb.org, accessed on 15 January 2023) [7].

The transmembrane (TM) domains of GPCRs are highly similar in architecture in spite of their sequence similarity, which could be as low as 20% [8]. According to the current theory, this domain controls the transmission of external signals to the intracellular surface of the protein. The extra- and intracellular loops and the extracellular (N-terminal) and cytosolic (C-terminal) domains are more diverse in terms of length and sequence similarity, and generally have more elastic structures. These domains have been proposed to be responsible for ligand and G protein/arrestin specificity. The number of G protein and arrestin subtypes is approximately two orders of magnitude lower than that of GPCRs. This and the remarkable three-dimensional similarity of TM domains suggests that the activation of GPCRs follows a general structural mechanism. The current theory of activation states that GPCRs may exist in multiple dynamically interconverting active and inactive structural states. Such different structural and functional states are emerging from the reversible rearrangement of the 3rd, 5th and 6th helices of the TM domain and most notably the horizontal displacement of the 6th TM helix [9]. The populations of these simultaneously occupied states are controlled by the bound ligand and intracellular signaling proteins [9]. The rearrangement of TM helices is generally accompanied by synchronous rearrangements of specific intramolecular interactions involving the ortho- and allosteric binding sites, conserved polar functional motifs (E/DRY, NPxxY, CWxP) and a network of water molecules in the inner cavities of the TM domain [10,11,12,13,14]. These sites and motifs form a continuous polar network connecting the orthosteric ligand binding pocket to the intracellular G protein binding surface, and this network has been proposed recently as the general machinery of receptor activation [15,16].

Previously, to provide further insight, we performed extensive MD simulations of active and inactive state μ-opioid receptor (MOP), bound by an endogenous agonist and by either the G_i_-protein heterotrimeric complex or β-arrestin-2. Our results indicated that the constituents of the above-mentioned polar signaling channel, connecting the ligand binding pocket to the intracellular surface, are engaged in highly correlated internal motions [17]. However, these concerted motions were only observed in the case of the G_i_ protein-bound active state MOP, suggesting that this phenomenon could be associated with G protein-mediated signaling. Similar trends were observed in our follow-up study involving the β_2_-adrenergic receptor (β_2_AR), providing further reinforcement to our previous assumptions while corroborating the above-cited independent proposals [18]. It was also shown that the polar signaling channel could be subdivided into two segments. The first segment, spanning the orthosteric binding pocket and the NPxxY motif, was proposed to be responsible for ligand-induced effects, while the concerted motions of the second segment furnish constitutional activity with the involvement of the intracellular tip of TM7 and residues of H8 [18]. Considering the fact that the function of class A GPCRs is dramatically affected by mutations that alter the polarity of conserved motifs [19,20,21,22,23,24,25,26,27,28], and sodium binding in the TM domain also inhibits the activation of numerous class A GPCRs [14], we proposed that a shift in charge balance takes place during receptor activation, propagated by the rearrangements of polar amino acid side chains in the central duct of the TM domain, leading to a change in polarity on the intracellular surface and eventual G protein dissociation.

An important challenge in the field of GPCR structural biology is to find explanations for the incidents when ligands with similar structures, physico-chemical properties and receptor binding affinities demonstrate markedly different functional properties. In parallel, it is important to elucidate which ligand properties and/or receptor states are involved in balanced or biased signaling. This latter question was successfully addressed by a recent study involving the angiotensin II type 1 receptor (AT_1_R), where different intracellular conformations of the receptor were revealed, depending on which pathway is predominantly activated. The receptor structures associated with G_q_ protein and β-arrestin-mediated signaling were termed as ‘canonical’ and ‘alternative’ conformations, respectively. The two different active states were defined by specific pairwise residue–residue distances and side chain conformations [29]. This was later corroborated by a follow-up study involving the MOP receptor [30]. It is important to mention that the ‘alternative’ conformations were only observed in long time-scale molecular dynamics (MD) simulations and not in the structures acquired using X-ray crystallography and cryo-electronmicroscopy (cryo-EM), possibly due to the applied experimental conditions that locked the receptors in their ‘canonical’ active states. This indicates the exceptional capacity of MD simulations to offer predictive insights into structural properties that are unattainable by conventional experimental techniques.

Here, an extensive, unbiased, atomistic MD simulation study of the full sequence CB1 receptor is presented, where the receptor is embedded in a native-like caveolar membrane bilayer in its active [31] or inactive state [32], bound by an endogenous agonist 2-arachidonoglycerol (2-AG) and either the heterotrimeric G_i_ protein or β-arrestin-2. The endogenous agonists of CB1, including 2-AG, are neutral and possess considerable lipophilic character, opposed to the endogenous agonists of the previously investigated MOP and β_2_AR systems, which are generally hydrophilic and contain at least one ionizable group. The analysis of MD trajectories aimed to examine if previous assumptions about the structural mechanism of activation and pathway specificity could be extended to the CB1 receptor. More specifically, we intended to test if the theory of shifting electrostatic balance in the TM region is plausible even in CB1, where, unlike in MOP and β_2_AR, no ionic interaction is involved in ligand binding.

## 2. Results and Discussion

### 2.1. Simulation System Integrity

Following the approach of our previous studies involving the MOP and β_2_AR receptors, simulation systems were built by attaching approximate structures of the N- and C-terminal domains to the experimentally derived TM domains of the CB1 receptor (see Methods) [17,18]. The purpose of the inclusion of these highly variable and flexible domains was to account for their effect on the dynamics of the TM helices, primarily exerted by their mass. These domains are generally missing from the experimental structures of GPCRs and, therefore, omitted from the corresponding MD simulation studies. Here, their inclusion was regarded to be specifically important, considering that the disposition of TM helices was proposed to have a central role in receptor activation [9,15,16]. Partial unfolding of N- and C-terminal domains was frequently observed during simulations, as indicated by the evolution of radii of gyration (Appendix A). However, the minimum distance between these domains never fell below 0.8 nm, hence possible artifacts emerging from artificial contacts between neighboring periodic images of the terminal domains could be excluded (Appendix A).

The orientation of the ligand in the binding pocket was found to be crucial in our earlier investigations of the β_2_AR. [18] In the default of the high-resolution experimental structure of the CB1 receptor complexed by the endogenous CB1 agonist 2-AG, the orientation of 2-AG in the orthosteric binding pocket of CB1 was modeled utilizing blind molecular docking. 2-AG was located in the binding pocket in the majority of the docking poses with low binding free energy. The pose used later as the starting structure for the MD simulations was selected on the basis of similarity to the crystallographic structure of CB1 bound by AM11542, a synthetic phytocannabinoid-derived agonist (Appendix A). No dissociation of 2-AG from the binding pocket of CB1 was observed in any of the simulations, but the ligand–receptor complex was notably dynamic, as was indicated by the evolution of RMSD values of the heavy atoms of 2-AG (Appendix A). Nevertheless, there is at least one MD trajectory for each simulation system, in which the ligand has maintained its initial position and orientation for the majority (>95%) of the simulation time. Trajectories with notable ligand displacement (2-AG-active CB1-G_i_ protein complex 2nd replica, 2-AG-active CB1-β-arrestin-2 complex 2nd replica, 2-AG-inactive CB1-β-arrestin-2 complex 1st replica) were not excluded from analysis, but the results were interpreted accordingly. The relative positions of the macromolecular components were stable during simulations and no dissociation was observed.

Na^+^ penetration into the allosteric Na^+^ binding site (D163^2.50^) was observed in two cases, the 2-AG-bound inactive CB1-G_i_ protein complex 2nd replica and the 2-AG-inactive CB1-β-arrestin-2 complex 1st replica. (Appendix A) No crystallographic coordinates of Na^+^ binding in the allosteric binding pocket of the CB1 receptor have been provided so far, but the allosteric effect of Na^+^ on ligand binding, controlled by D163^2.50^, has been evidenced earlier [33]. Even though Na^+^ binding has been shown to have less dramatic effect on ligand binding to the CB1 receptor than to other class A GPCRs, Na^+^ penetration observed during our inactive CB1 simulations is a likely outcome. However, it is interesting to note that Na^+^ penetration did not always coincide with ligand displacement (Table 1, Appendix A). In previous simulations of the MOP and β_2_AR receptors, sodium penetration was only observed when the orthosteric pocket was unoccupied, suggesting that the allosteric Na^+^ binding site of class A GPCRs is only accessible through the orthosteric pocket and the bound orthosteric ligand blocks the entrance of Na^+^ [17,18]. The results obtained for the 2-AG-bound inactive CB1-G_i_ protein complex, where the position of 2-AG was relatively stable in the orthosteric site throughout the course of simulation, suggest that the allosteric site of this receptor is more readily accessible from the extracellular side. Na^+^ entrance from the cytosolic side did not occur in any of the systems; therefore, intracellular access of Na^+^ ions through the TM domain is most likely to be obstructed by the intracellular signaling proteins.

### 2.2. Transmembrane Helix and Loop Dynamics

Analysis of the dynamics of the 6th transmembrane helix (TM6), the hallmark conformational switch of class A GPCR activation, indicated small to moderate displacements of this helix from the corresponding starting structures in all simulation setups, and no complete transitions between active and inactive receptor conformations were observed (Figure 1). This is similar to our previous simulation results for the MOP and the β_2_AR, where larger dispositions of TM6 were only observed in the absence of ligands, suggesting that bound agonists may have a remarkable stabilizing effect on the structure of the TM domain. Transitions between functional states were observed previously in MD simulations of the β_2_AR, but on significantly longer timescales and in the absence of bound intracellular proteins [34]. Here, the internal motions of TM6 were observed to depend on three major factors: the initial functional state of the receptor (active vs. inactive starting structure), the stability of the receptor–ligand complex and the bound intracellular signaling protein (Table 1). While ligand disposition was fairly tolerated in the active G_i_ protein-bound CB1, it resulted in notable TM6 destabilization in β-arrestin-2 complexes. Furthermore, TM6 position was maintained in inactive state systems in which 2-AG was snugly bound.

In our previous study, large dispositions (~0.4 nm RMSD) of the conserved NPxxY motif in the 7th transmembrane helix (TM7) were found to coincide with intensive concerted dynamics of the second segment of the polar signaling channel (closer to the intracellular surface), suggesting that the mobility of this segment could be associated either with receptor activation or constitutional activity [18]. In the CB1 simulations, such large dispositions of the NPxxY domain were found to rather coincide with intensive ligand movements and disposition, which, in contrast with our previous observations, apparently resulted in the decoupling of correlated side chain motions in the polar signaling channel (see further discussion below) (Figure 2, Table 1). This, similar to the results presented above for TM6 disposition, implies the interplay of multiple structural factors, states and conditions in the activation mechanism.

Secondary structure analysis of the 1st, 2nd and 3rd intracellular loops (ICL1, ICL2 and ICL3, respectively) and the cytosolic helix (H8) indicated that these molecular parts maintain highly similar structures in all receptor states, and only minor or random differences could be observed, emerging from either the starting receptor conformations (H8) or high inherent flexibility (ICL1) (Appendix A). The results of secondary structure analysis of TM7 suggest a higher degree of mobility of the intracellular tip of TM7 in the active states, which could be associated either with receptor activation or constitutional activity, but such an association is not directly corroborated by the NPxxY dynamics data above (Figure 2, Table 1, Appendix A).

### 2.3. Intramolecular Interactions

Table 2 reports the results of salt bridge and H-bond analysis. The frequency data represent intermolecular interactions that were proposed previously to be important for class A GPCR activation. No interaction between the DRY motif and H8 could be found in any of the systems. The presence of a specific salt bridge between R165^3.50^ of the DRY motif and D340^8.47^ of H8 was first indicated in our earlier MD simulation study of the MOP [17]. The presence and specificity of this interaction was evidenced recently, and it was suggested that this interaction is particularly important for stabilizing an alternative receptor conformation that facilitates β-arrestin-2 recruitment to the MOP [30]. However, this interaction was also found to be missing in our earlier study of the β_2_AR; therefore, it is most likely a specific property of the MOP receptor and irrelevant for CB1. [18] In agreement with experimental data, the frequent occurrence of salt bridges and H-bonds were observed between the D213^3.49^ and R214^3.50^ residues of the DRY motif in the inactive states. These interactions were also observed, although with lower frequencies, in the active CB1–β-arrestin-2 complex. Such an interaction was also indicated in the cryo-EM structure of the active β_1_-adrenergic receptor (β_1_AR)–β-arrestin-1 complex, but it was missing from the neurotensin-1 receptor (NTS_1_R) bound by β-arrestin-1 [35,36]. The presence of an ‘ionic lock’ interaction between the DRY motif and TM6, observed earlier in the X-ray crystallographic structure of rhodopsin, was corroborated by our simulation results [11]. The frequent occurrence of a salt bridge and/or an H-bond between residues R213^3.50^ and D338^6.30^ was indicated in the trajectories obtained for the inactive state CB1. This interaction was proposed to act as a constraint in the inactive state, which is disrupted upon receptor activation, allowing TM6 to move outward. In agreement with the experimental structures, H-bonds were systematically present between D213^3.49^ of the DRY motif and Y224 of ICL2 in both the active and inactive CB1, regardless of the type of bound intracellular signaling protein [31,32]. The interaction between the DRY motif and ICL2 was shown previously to reorganize upon the activation of the β_2_AR [17,37]. Taken together the results of secondary structure and intramolecular interactions analysis, no similar trends could be deduced for the CB1 receptor. Additionally, no correlation was found between the different receptor states and the frequencies of the CWxP-TM7 interaction within the time frame of simulations [20,38]. However, the DRY-TM5 interaction, proposed earlier to stabilize the G protein-bound active state, was confidently reproduced in the CB1 simulation systems [39].

### 2.4. Correlated Side Chain Motions in the Transmembrane Domain

Cross-correlation analysis (Figure 3 and Appendix A) of the amino acid side chain dynamics in the TM domain indicated that, similar to our earlier observations taken for the MOP and β_2_AR receptors [17,18], the orthosteric binding pocket and the intracellular surface of the CB1 receptor are connected through a channel of conserved polar amino acid residues that are engaged in concerted motions in the relevant active signaling states (Figure 4 and Appendix A). The unique feature of the polar signaling channel of the CB1 receptor identified here is that it is activated in β-arrestin-2-bound states too, and not solely in the active G protein-bound state as observed for the MOP and β_2_AR receptors. This is an astounding result, considering that 2-AG has been reported previously as a G protein-biased agonist [6]. It is important to point out that these correlated motions are not a mere consequence of increased conformational mobility in the TM region. The receptor has to maintain a specific degree of order and stability in order for these concerted motions to take place. This is clearly indicated by the results presented above, where apparent structural destabilization in the binding pocket and consequently near the NPxxY motif led to the decoupling of concerted motions of the polar signaling channel. Further support is provided by earlier simulations of ligand-free MOP and β_2_AR receptors, where the absence of ligands resulted in an elevated conformational flexibility of the TM domain and a complete loss of correlated motions of the polar signaling channel residues [17,18]. It is also important to note that correlated motions were more affected in the second segment of the polar signaling channel, close to the intracellular surface (Table 1). This suggests that the observed effects emerge from direct interactions with the different intracellular signaling proteins applied in this study. In terms of the involvement of conserved residues and motifs, the polar signaling channel of the CB1 receptor is found to be more similar to that of the β_2_AR receptor than to that of the MOP (Figure 4). In the active, G_i_ protein-bound MOP, the DRY motif is coupled to the polar signaling channel through a salt bridge between R165^3.50^ (DRY) and D340^8.47^ (H8). In β_2_AR and CB1, the analogous residues in H8 are S329^8.47^ and S401^8.47^, respectively, which cannot form a salt bridge. The frequency of H-bonds between the above residues of H8 and the DRY motif of CB1 were also found to be negligible (see above). Therefore, the contribution of the DRY motif to the polar signaling channel could be a specific property of the MOP receptor, as discussed above. Similar to β_2_AR, H8 of CB1 was found to be engaged in the correlated motions of the signaling channel by contributing residues D403^8.49^ and R405^8.51^ (Figure 4), among which R405^8.51^ is conserved (Figure 4), and was implicated previously in G protein coupling to the adenosine A_2B_ receptor [21]. A further similarity between the polar signaling channels of the CB1 and β_2_AR receptors and the difference to that of the MOP is that the allosteric Na^+^ binding site (D163^2.50^ in CB1) is less involved in the concerted motions of channel residues. In β_2_AR, the coupling of D79^2.50^ to the signaling cascade was restored when epinephrine was mildly restrained to the orthosteric binding pocket, suggesting the crucial importance of strong binding and correct orientation of the ligand. [18] Here, in the case of CB1, this lack of involvement is most likely due to the fact that the physico-chemical features of the allosteric and orthosteric binding pockets and the respective ligands are significantly different (polar, charged vs. hydrophobic/lipophilic), which may suggest that the connection between the ortho- and allosteric sites is not as pronounced as in other class A GPCRs. Furthermore, the allosteric site and the residues of the signaling channel along TM7 are apparently more easily accessible to Na^+^ than in other class A GPCRs (see above); therefore, conformational transitions of D163^2.50^ could be less important.

### 2.5. Alternative Signaling States

The analysis of specific pairwise residue–residue distances and side chain conformations in the TM domain, which were reported previously to be different for G_i_ protein- and β-arrestin-2-mediated signaling (Appendix A), indicated that the starting structures of the active state MD simulations were canonical, favored for G_i_ protein binding. The canonical structural state was maintained throughout the simulations of the G_i_ protein-bound active CB1 receptor, although several notable differences were observed. The distance between residues N134^1.50^ and S390^7.46^ fell in the range of the alternative active states for all systems, and only a temporary transition to the canonical state was observed in one of the β-arrestin-2 complex simulations. The distances between F170^2.57^ and F200^3.36^ and between N134^1.50^ and T391^7.47^ also differed from those reported previously for the canonical active state, but the relative orientations of the corresponding residues were still close to the former, rather than to the alternative state. The side chain conformation of Y397^7.53^ was found to be slightly different from *trans*, corresponding to the alternative state, during the 2nd replica simulation of the active CB1–G_i_ protein complex. The *gauche-* conformation was expected for this residue in this complex, corresponding to the canonical signaling state. Nevertheless, this observation may have less relevance than the above differences, considering that this simulation was divergent in terms of system integrity.

In the trajectories of the CB1–β-arrestin-2 complexes, the transition to the alternative structural state, favored for β-arrestin-2 binding, was observed (Table 1 and Table 3, Appendix A). However, as the results indicate, these transitions were partial and incomplete. They took place in the structural region close to the intracellular surface, while the receptor structure closer to the orthosteric binding pocket was preserved throughout the simulations (Table 3, Appendix A). The apparent structural destabilization close to the intracellular surface is a possible outcome of β-arrestin-2 binding to the initial canonical state CB1 receptor, while such structural transition around the orthosteric pocket was prevented by the loosely bound G_i_ protein-biased ligand. Presumably, a complete transition from the canonical to the alternative active state would require the presence of a β-arrestin-2-biased agonist, as well as a longer simulation time. Nevertheless, the results presented here are satisfactory to support the proposal about biased signaling through multiple active structural states [29,30].

## 3. Conclusions

The results obtained for the CB1 receptor and presented here are sufficient to support our polar signaling channel model of class A GPCR signaling and are in agreement with independent proposals [11,15,16,17,29,30,35,36]. Apart from similarities to other class A GPCRs, specific features of the CB1 receptor have been revealed, including the active polar signaling channel in β-arrestin-2-bound, semi-active states and the accessibility of the allosteric Na^+^ binding pocket. CB1 is the third class A GPCR in which the role of correlated motions of conserved polar motifs have been indicated, which suggests a potential contribution of the electrostatic balance in the TM domain. Further investigations involving specific receptor mutations, rationally designed ligand probes and in-depth quantitative analysis of the electrostatic balance utilizing mixed quantum mechanical/molecular mechanical methods could valuably contribute to the current knowledge about the structural and physico-chemical features of CB1-mediated signaling. The general features of receptor activation uncovered, as well as receptor-specific characteristic details, may find use in the discovery and development of a new class of GPCR drugs devoid of harmful side effects. 

A holistic view on the activation mechanism that includes multiple perspectives could provide explanations for how different functional states are induced and what specific physico-chemical properties of GPCR ligands are responsible.

## 4. Methods

### 4.1. System Building

The sequence of the CB1 receptor (UniProtKB-P21554-CNR1) was obtained from the UniProt database (http://www.uniprot.org, accessed on 2 February 2021). The cryo-EM structure of the active state CB1–G_i_ protein complex (pdb code: 6N4B) [31] and the X-ray crystallographic structure of the inactive state CB1 (pdb code: 5TGZ) [32], as well as separate coordinates of the G_i_ protein heterotrimeric complex and GDP (pdb code: 1GP2) [40] and β-arrestin-2 (pdb code: 3P2D) [41], were downloaded from the Brookhaven Protein Data Bank (http://www.rcsb.org, accessed on 2 February 2021). The ligand used for the study is 2-arachidonoglycerol (2-AG), which, along with the above structures, was included in the starting structures of the MD simulations. A considerable portion of the G_α_ subunit is missing from the active state CB1 cryo-EM structure (pdb code: 6N4B) [31], hence coordinates of this particular subunit were supplemented from the separate crystallographic structure of the G_i_ heterotrimeric complex (pdb code: 1GP2) [40]. The crystallographic structures of the G_s_ protein-bound active β_2_AR (pdb code: 3SN6) [38] and the visual arrestin-bound rhodopsin (pdb code: 4ZWJ) [42] served as templates for the alignment of G_i_ protein- and β-arrestin-2-bound CB1 complexes, respectively. The orientation of 2-AG was obtained from blind docking of the ligand to the active state receptor (pdb code: 6N4B) [31] using the Autodock ver. 4.2 software [43] and the Lamarckian genetic algorithm. All ϕ, ψ and χ^1^ ligand torsions, as well as receptor side chains previously indicated to be involved in ligand–receptor interactions, were kept flexible. [44] The docking of 2-AG was performed in an 8.0 nm × 8.0 nm × 8.0 nm grid volume, large enough to cover the whole binding pocket of the receptor region accessible from the extracellular side. The spacing of grid points was set at 0.0375 nm and 1000 dockings were carried out. The resultant ligand–receptor complexes were clustered and ranked according to their corresponding binding free energies. The lowest energy bound state conforming to the ligand orientation observed in the CB1—AM11542 experimental complex (pdb code: 5XRA) [44] was selected for the starting structure of the simulations. All crystallization chaperones or fusion proteins present in experimental CB1 structures were removed to ensure the specificity of the results, and to ensure accurate and complete representation. The missing third intracellular loop (ICL3) of active (Q314-Q334) and inactive CB1 (R307-R331), and the second extracellular loop (ECL2) of active CB1 (E258-V263) absent from the experimental structures were modeled using the Modeller ver. 9.20 software [45]. Other missing or mutated residues in the structures were incorporated or restored using the Swiss-PdbViewer program (ver. 4.10) [46]. The missing N- and C-terminal segments of CB1 (M1-M103 and C415-K472, respectively) were reconstructed via 10 ns folding simulations utilizing the GROMACS ver. 2018.3 software package [47], in accordance with a previously established protocol [17,18], and subsequently manually appended to the TM domain of the receptor.

The CHARMM-GUI web-based platform was utilized to integrate the post-translational modifications of CB1 and to construct a solvated membrane bilayer in which the receptor was embedded [48]. Specifically, the N-terminal domain of the receptor was glycosylated at residues N77 and N83 [49,50,51,52]. Complex type glycans were employed for the glycosylation of the N-terminal domain, which consisted of a common core (Manα1–3 (Manα1–6) Manβ1–4GlcNAcβ1–4GlcNAcβ1–N) and sialic acid (N-acetylneuraminic acid). In addition to that, residue C415 was palmitoylated in both the active and inactive states. [53] Furthermore, phosphorylation at the C-terminal domain (T460, S462, S464, T465, T467 and S468) was included only for systems bound to β-arrestin-2 [54,55].

The receptor complexes were placed into an asymmetric caveolar membrane bilayer that has been previously used and validated [17]. Briefly, the bilayer was composed of 32.8% cholesterol (CHL), 27.8% 1-palmitoyl-2-oleoyl-sn-glycero-3-phosphoethanolamine (POPE), 14.9% 1-palmitoyl-2-oleoyl-glycero-3-phosphocholine (POPC), 6.0% 1-palmitoyl-2-oleoyl-sn-glycero-3-phosphoinositol (POPI2), 3.6% 1-palmitoyl-2-oleoyl-sn-glycero-3-phospho-L-serine (POPS), 5.0% monosialodihexosylganglioside (GM3) and 9.9% palmitoyl-sphingomyelin (PSM) [56]. The asymmetric composition of the lower and upper leaflets was determined according to recent literature data [57]. The CB1 complexes were then inserted into this membrane using the CHARMM-GUI membrane builder tool [48]. The system was surrounded by explicit TIP3P water molecules in a hexagonal periodic box, and Na^+^ and Cl^−^ ions were added at 0.15 M concentration to make the system electrically neutral and match the ionic strength in physiological systems. The system coordinates and topologies were generated in GROMACS format, and CHARMM36 all-atom force field parameters were assigned to all the system components. [58]

### 4.2. MD Simulations

The GROMACS 2018.3 program package was used to perform energy minimization and MD simulations of the complexes [47]. Simulation systems were subjected to an initial 5000 steps of steepest descent, followed by 5000 steps of conjugate gradient energy minimization, with a convergence criterion of 1000 kJ/mol/nm for both steps. The minimized systems were then equilibrated using a six step protocol supplied by CHARMM-GUI, which consisted of two consecutive MD simulations at 303.15 K in the canonical (NVT) ensemble, followed by four successive simulations at 303.15 K and 1 bar pressure in the isobaric-isothermic (NPT) ensemble. The equilibration protocol included the application of positional restraints on the heavy atoms of the proteins and membrane constituents, which were gradually decreased throughout the steps. The first three equilibration MD runs were 25 ps in length and performed with 1 fs time steps. The next two runs were continued for 100 ps in 2 fs time steps, and the final equilibration step was extended to 50 ns, executed in 2 fs time steps. The LINCS algorithm was used to constrain chemical bonds to their correct lengths, the v-rescale algorithm [59] with a coupling constant of 1 ps was used to regulate temperature and the Berendsen (semi-isotropic) pressure coupling [60] method with a 5 ps coupling constant and an isothermal compressibility of 4.5 × 10^−5^ bar^−1^ was applied for pressure control. The particle mesh Ewald (PME) summation was used to calculate energy contributions from long-range electrostatic interactions, and a twin-range cutoff was used to calculate van der Waals interactions. All cut-off values were set to 1.2 nm. In order to study the dynamic behavior of the active and inactive state CB1 bound to orthosterically bound 2-AG, and complexed either with the heterotrimeric G_i_ protein or β-arrestin-2, eight independent production simulations were performed at 310 K in the NPT ensemble. Each system was simulated in two replicates, yielding 8 μs of trajectories across all simulations. The system coordinates were stored in every 5000 steps, providing trajectories with 100,000 snapshots.

### 4.3. MD Trajectory Analysis

The analysis of the MD trajectories was performed using the GROMACS 2018.3 package analysis suite [47]. A set of specific analyses was conducted to compare the results with our previous works on MOP and β_2_AR receptors [17,18]. The gmx rms tool was used for the calculation of root mean square deviation (RMSD) of protein backbone atoms to evaluate the structural stability of the macromolecular complexes and to identify significant displacements of key structural components. The radii of gyration of terminal domains were calculated using gmx gyrate to assess their structural flexibility during simulations. Potential artificial contacts between periodic replicas of the terminal domains were checked for using gmx mindist. The gmx mindist tool was also employed to observe Na^+^ penetration into the allosteric sodium binding site, D163^2.50^. The DSSP method was used to track the evolution of the secondary structure of different domains [61]. The gmx hbond utility was used to determine the frequency of intermolecular H-bonds. These bonds were assigned with a donor–acceptor distance cut-off of 0.35 nm and a donor–hydrogen-acceptor angle of 30.0 degrees or below. The gmx distance and gmx gangle utilities were used to identify salt bridges between acidic and basic functional groups with cutoff values of 0.4 nm for the distance and 90.0 for the angle between the participants. The evolution of specific residue–residue distances and side chain rotamer conformations defining different signaling states was analyzed using the gmx distance and gmx chi utilities of the GROMACS 2018.3 suite, respectively. The dynamics of amino acid side chains in the transmembrane domain and connecting loops were inspected through generalized cross correlation matrix analysis computed on the basis of linear mutual information (GCC-LMI), integrated into an earlier version of GROMACS (g_correlation, GROMACS 3.3) [62]. The correlation matrices were converted to heat maps using the gmx xpm2ps utility, and Gimp (version 2.10.30) software was employed for subsequent analysis. The threshold for correlation assignment was set at red color intensity corresponding to >0.63 mutual information (MI) values, and the participation of at least four atoms from each amino acid side chain was considered. Molecular visualizations were aided by VMD (ver. 1.9.4a12) [63] and Pymol (ver. 2.6.0a0), and graphs were created using the Xmgrace (ver. 5.1.25) program.

### 4.4. Sequence Alignment and Conservation Analysis

A dataset of 267 sequences of class A human GPCRs was obtained from the UniProt database in FASTA format. The dataset excluded orphan and olfactory receptors. Multiple-sequence alignment of the dataset was performed using the Clustal Omega program. [64] The alignment was subsequently analyzed using the Jalview 2.10.5 software [65]. The CNR1_HUMAN (P21554) sequence was selected as the reference sequence for the conservation analysis based on the percentage of identity.

## Figures and Tables

**Figure 1 ijms-24-04232-f001:**
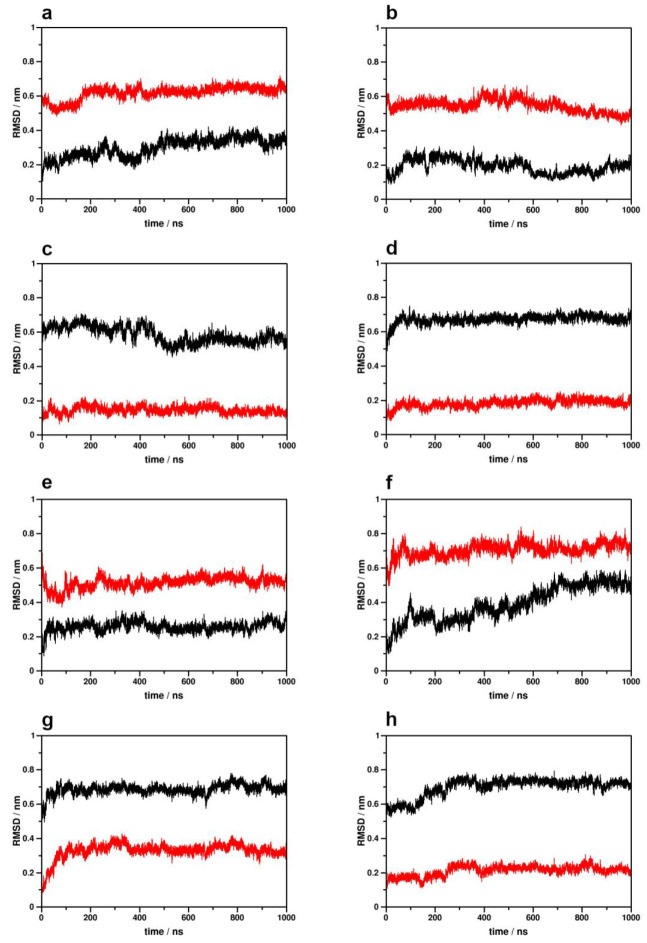
Disposition of the TM6 helix during simulations with respect to the active (black) and inactive (red) crystallographic structures of the CB1 receptor. (**a**) Active CB1–G_i_ protein complex, 1st replica; (**b**) active CB1–G_i_ protein complex, 2nd replica; (**c**) inactive CB1–G_i_ protein complex, 1st replica; (**d**) inactive CB1–G_i_ protein complex, 2nd replica; (**e**) active CB1–β-arrestin-2 complex, 1st replica; (**f**) active CB1–β-arrestin-2 complex, 2nd replica; (**g**) inactive CB1–β-arrestin-2 complex, 1st replica; (**h**) inactive CB1—β-arrestin-2 complex, 2nd replica.

**Figure 2 ijms-24-04232-f002:**
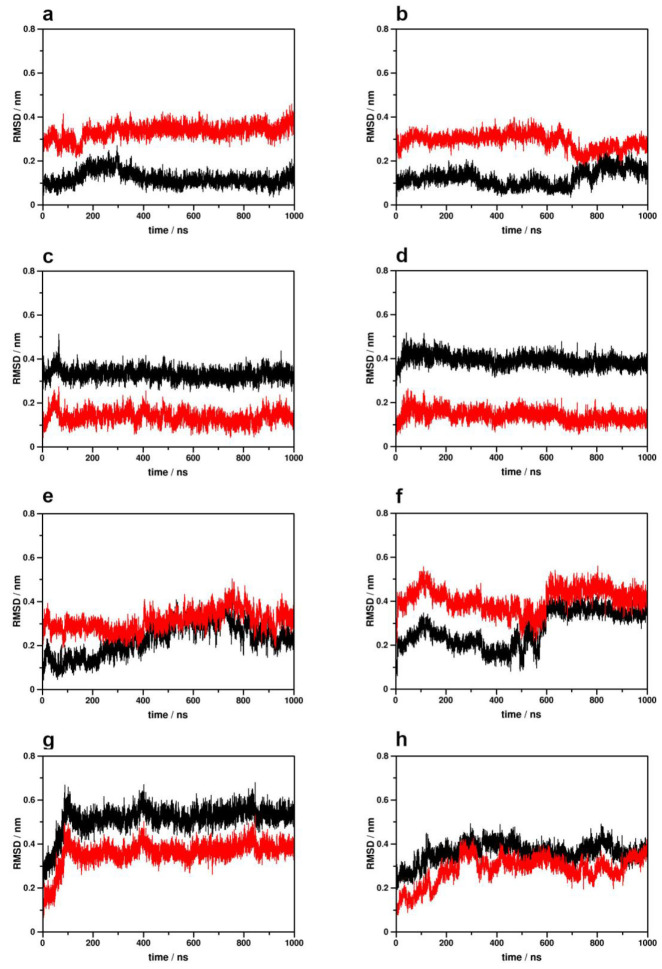
Disposition of the NPxxY motif during simulations with respect to the active (black) and inactive (red) crystallographic structures of the CB1 receptor. (**a**) Active CB1–G_i_ protein complex, 1st replica; (**b**) active CB1–G_i_ protein complex, 2nd replica; (**c**) inactive CB1–G_i_ protein complex, 1st replica; (**d**) inactive CB1–G_i_ protein complex, 2nd replica; (**e**) active CB1–β-arrestin-2 complex, 1st replica; (**f**) active CB1–β-arrestin-2 complex, 2nd replica; (**g**) inactive CB1–β-arrestin-2 complex, 1st replica; (**h**) inactive CB1–β-arrestin-2 complex, 2nd replica.

**Figure 3 ijms-24-04232-f003:**
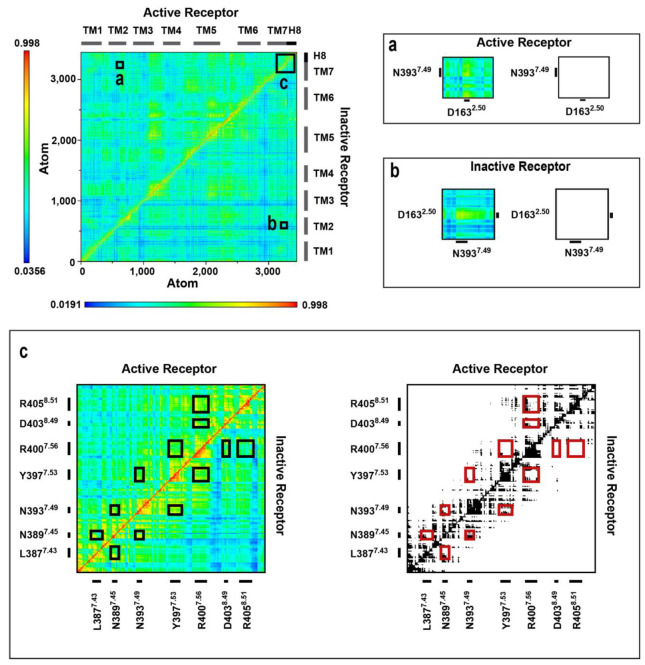
Cross-correlation matrices of the G_i_ protein-bound CB1 (1st replica) in the active and inactive states. Panels (**a**–**c**) are magnified views of regions of amino acid residues of interest. Black and white panels show correlations above the threshold of 0.63 MI.

**Figure 4 ijms-24-04232-f004:**
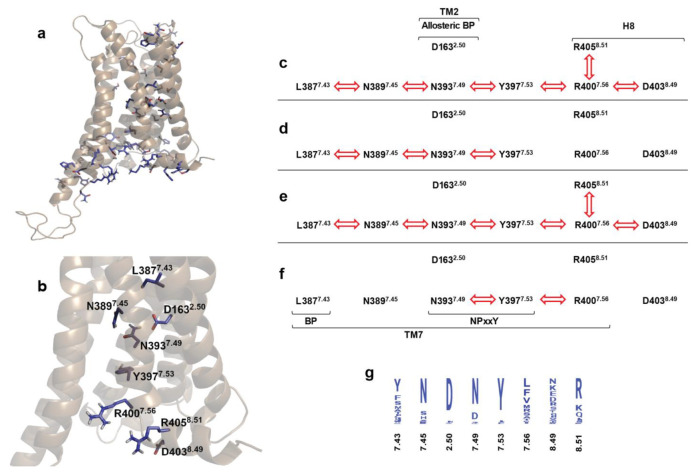
The polar signaling channel of the CB1 receptor identified through cross correlation analysis. (**a**) Polar amino acids of which motions are correlated in the G_i_ protein-bound active state. (**b**) Polar amino acids of which motions are correlated and connecting the orthosteric binding pocket to the G_i_ protein-binding interface. Non-polar hydrogens are omitted for clarity. (**c**) Active CB1–G_i_ protein complex, 1st replica; (**d**) inactive CB1–Gi protein complex, 2nd replica; (**e**) active CB1–β-arrestin-2 complex, 1st replica; (**f**) inactive CB1–β-arrestin-2 complex, 2nd replica. Red arrows indicate correlated motions of the respective amino acids. (**g**) Degree of conservation of polar signaling channel residues of human class A GPCRs.

**Table 1 ijms-24-04232-t001:** Summary and comparison of major observations taken during the simulations of the CB1 receptor complexes.

	Ligand Disposition	Na^+^ Penetration	TM6 Disposition	NPxxY Disposition	Correlated Motions in Segment 1	Correlated Motions in Segment 2	Signaling State
**active CB1-G_i_ complex, replica 1**	moderate	no	moderate	stable	complete	complete	canonical
**active CB1-G_i_ complex, replica 2**	large	no	stable	transition to intermediate	complete	complete	canonical
**inactive CB1-G_i_ complex, replica 1**	moderate	no	stable	stable	complete	incomplete	N/A
**inactive CB1-G_i_ complex, replica 2**	moderate	yes	stable	stable	complete	incomplete	N/A
**active CB1-β-Arr-2 complex, replica 1**	moderate	no	moderate	transition to intermediate	complete	complete	transition to alternative
**active CB1-β-Arr-2 complex, replica 2**	large	no	transition to inactive	transition to intermediate	complete	incomplete	transition to alternative
**inactive CB1-β-Arr-2 complex, replica 1**	large	yes	transition to intermediate	transition to intermediate	complete	incomplete	N/A
**inactive CB1-β-Arr-2 complex, replica 2**	moderate	no	stable	transition to intermediate	incomplete	incomplete	N/A

N/A = not applicable.

**Table 2 ijms-24-04232-t002:** Frequency of intramolecular salt bridges and H-bonds expressed as percentages of the total conformational ensemble, generated by MD simulations.

Interactions	Residues Involved	G_i_ Protein Complex				β-Arrestin-2			
		Active State		Inactive State		Active State		Inactive State	
		Replica 1	Replica 2	Replica 1	Replica 2	Replica 1	Replica 2	Replica 1	Replica 2
**salt bridges**									
**DRY—H8**	R214^3.50^; D403^8.49^	0.0	0.0	0.0	0.0	0.0	0.0	0.0	0.0
**intra-DRY**	D213^3.49^; R214^3.50^	0.0	0.0	25.6	25.3	24.4	3.6	27.33	27.24
**DRY—TM6**	R214^3.50^; D338^6.30^	0.0	0.0	16.5	30.5	0.0	0.0	17.2	26.1

**H-bonds**									
**DRY—H8**	R214^3.50^; S401^8.47^	0.0	0.0	0.0	0.0	0.0	0.1	0.0	0.0
	R214^3.50^; D403^8.49^	0.0	0.0	0.0	0.0	0.0	0.1	0.0	0.0
	R214^3.50^; R405^8.51^	0.0	0.0	0.0	0.0	0.0	0.0	0.0	0.0
**intra-DRY**	D213^3.49^; R214^3.50^	0.1	0.0	99.9	99.8	94.9	20.5	99.8	99.9
**DRY—ICL2**	D213^3.49^; R220^ICL2^-T229^ICL2^	96.1	99.8	99.9	99.6	99.5	99.7	99.8	99.7
**DRY—TM5**	R214^3.50^; Y294^5.58^	96.3	64.4	0.0	0.0	0.1	0.2	0.0	0.0
**DRY—TM6**	R214^3.50^; D338^6.30^	0.0	0.0	37.8	98.8	0.0	0.0	43.5	99.6
**CWxP—TM7**	C355^6.47^-W356^6.48^; N389^7.45^	6.9	24.1	11.6	13.8	14.1	42.3	0.1	0.1

Ballesteros–Weinstein numbering of residues is indicated in superscript.

**Table 3 ijms-24-04232-t003:** Specific residue–residue distances and side chain conformations that characterize the canonical and alternative active states favored for G_i_ protein and β-arrestin-2-mediated signaling, respectively.

Pathway-Specific Distance/Dihedral Angle	Canonical Active State	Alternative Active State	Active CB1-G_i_ Complex, Replica 1	Active CB1-G_i_ Complex, Replica 2	Active CB1-β-Arr-2 Complex, Replica 1	Active CB1-β-Arr-2 Complex, Replica 2
**F170^2.57^—F200^3.36^**	~1.00 nm	~1.50 nm	canonical *	canonical *	canonical *	canonical *
**D163^2.50^—S199^3.35^**	~1.00 nm	~0.50 nm	canonical	canonical	canonical	canonical
**F170^2.57^—L387^7.43^**	~0.50 nm	~1.00 nm	canonical	canonical	canonical	canonical
**G166^2.53^—F170^2.57^**	~0.75 nm	~0.50 nm	canonical	canonical	canonical	canonical
**N134^1.50^—T391^7.47^**	~0.50 nm	~0.25 nm	canonical *	canonical *	canonical *	canonical *, large fluctuations
**N134^1.50^—S390^7.46^**	~0.25 nm	~0.60 nm	alternative	alternative	switching between the two states	alternative, large fluctuations
**L159^2.46^—P394^7.50^**	~0.79 nm	~0.64 nm	canonical	canonical	canonical, large fluctuations	switching between the two states
**Y397^7.53^ χ^1^**	*gauche-*	*trans*	canonical	alternative *	alternative	alternative

* Distance and dihedral angle values were observed to differ from those reported previously for the canonical and alternative active states of the AT_1_R receptor. Ballesteros–Weinstein numbering of residues is indicated in superscript.

## Data Availability

All data presented in the article and the Appendix A are available upon request from the corresponding author.

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
