# Peer review of "All-Atom Molecular Dynamics Simulations Indicated the Involvement of a Conserved Polar Signaling Channel in the Activation Mechanism of the Type I Cannabinoid Receptor"

_ijms, 2023, doi:10.3390/ijms24044232_

Round 1

Reviewer 1 Report

Title: Should be revised. It is not enough to represent what the authors investigated in the study.

Abstract: Authors should add a brief sentence about the importance of CB1 receptors.

Revise the sentence “The knowledge base of GPCR structure has been expanding quickly in the 13 past decade.”

Authors should avoid “Apart from the confirmation of general features 27 of the activation mechanism, several unique properties of the CB1 have been discovered which 28 could be associated with a signaling profile specific to this receptor.” too confident statements

Introduction

Authors should provide the basis of their research and the power of their method

Add reference for line 70

Authors should revise table 2.

Authors should expand discussion section and make suggestions for future studies based on their findings.

Author Response

Answers to Reviewer 1.

Thank you for your opinion and constructive recommendations. Please find our answers below, as well as our explanations and modifications made to the manuscript, following your suggestions.  We hope that our modifications led to sufficient improvement of the manuscript. Our answers are printed in bold and line numbers refer to those in the revised manuscript.

English language and style

( ) English very difficult to understand/incomprehensible
( ) Extensive editing of English language and style required
( ) Moderate English changes required
(x) English language and style are fine/minor spell check required
( ) I don't feel qualified to judge about the English language and style

Yes

Can be improved

Must be improved

Not applicable

Does the introduction provide sufficient background and include all relevant references?

( )

(x)

( )

( )

Are all the cited references relevant to the research?

(x)

( )

( )

( )

Is the research design appropriate?

(x)

( )

( )

( )

Are the methods adequately described?

(x)

( )

( )

( )

Are the results clearly presented?

( )

(x)

( )

( )

Are the conclusions supported by the results?

(x)

( )

( )

( )

+

Comments and Suggestions for Authors

Title: Should be revised. It is not enough to represent what the authors investigated in the study.

The title has been changed to be more informative by including the subject of the study, the applied method and the most important observation (line 2-4):

‘All-atom molecular dynamics simulations indicated the involvement of a conserved polar signaling channel in the activation mechanism of the type I cannabinoid receptor.’

Abstract: Authors should add a brief sentence about the importance of CB1 receptors.

In order to provide a more specific statement about the importance of the CB1 receptor the following sentence of the Abstract:

‘The type I cannabinoid G protein-coupled receptor (CB1, GPCR) is an intensely investigated pharmacological target.’

has been replaced by

‘The type I cannabinoid G protein-coupled receptor (CB1, GPCR) is an intensely investigated pharmacological target, owing to its involvement in numerous physiological functions as well as pathological processes such as cancers, neurodegenerative diseases, metabolic disorders and neuropathic pain.’ (line 11-14)

Revise the sentence “The knowledge base of GPCR structure has been expanding quickly in the 13 past decade.”

The above sentence has been replaced by the following:

‘The pool of atomic resolution experimental structures of GPCRs has been expanding rapidly in the last decade, providing invaluable information about the function of these receptors.’ (line 16-17)

Authors should avoid “Apart from the confirmation of general features 27 of the activation mechanism, several unique properties of the CB1 have been discovered which 28 could be associated with a signaling profile specific to this receptor.” too confident statements

The above sentence was reiterated as follows:

‘Apart from the identification of previously proposed, general features of the activation mechanism, several specific properties of the CB1 have been indicated which could possibly be associated with the signaling profile of this receptor.’ (line 31-33)

Introduction

Authors should provide the basis of their research and the power of their method

The Introduction section includes a paragraph (lines 83-103) with references which describes the previous studies and results which have provided the specific basis and motivation for this present study of the CB1 receptor, including a hypothesis of which plausibility was meant to be tested by this present study. The next paragraph (line 104-119) presents the most recent results about structural indicators of signaling pathway specificity in GPCRs. These summarize the basis of our research and our specific aims are stated explicitly in the last two sentences of the Introduction. (line 128-133)

In consideration of the above, we think that the description of our research subject, aims and motivations are sufficiently detailed. Nevertheless, we would gladly make modifications and additions upon specific instructions. To further emphasize the power of the applied method, the following sentence was inserted in the Introduction section:

‘This indicates the exceptional capacity of MD simulations to offer predictive insights into structural properties of that are unattainable by conventional experimental techniques.’ (line 119-121)

Add reference for line 70 (line 74)

The citation of ref. 9 in line 76. was meant to underpin statements in line 70-76. Nevertheless, to avoid confusion, ref 9. İs now included in line 74 as well.

Authors should revise table 2.

We would gladly modify table 2 upon specific instructions or recommmendations from the Reviewer. Without such we could do no more than to fix formatting and typographic errors.

Authors should expand discussion section and make suggestions for future studies based on their findings.

We would like to reserve the Results and Discussion section for the presentation of results, their possible explanations and comparisons to previously published data and theories assumptions. To fulfill the request of the reviewer and to give more emphasis to potential future research on the basis of the findings of this study, we made the following modification to the Conclusion section:

‘CB1 is the third class A GPCR in which the role of correlated motions of conserved polar motifs have been indicated, which suggests potential contribution of the electrostatic balance in the TM domain, and the investigation could be extended to include in-depth quantitative analysis, utilizing quantum mechanical methods.’ (line 380-382)

Has been replaced by

‘CB1 is the third class A GPCR in which the role of correlated motions of conserved polar motifs have been indicated, which suggests potential contribution of the electrostatic balance in the TM domain. Further investigations involving specific receptor mutations, rationally designed ligand probes and in-depth quantitative analysis of the electrostatic balance utilizing mixed quantum mechanical / molecular mechanical methods could valuably contribute to the current knowledge about the structural and physico-chemical features of CB1-mediated signaling.’ (line 380-386)

Again, Thank you for your kind review and useful recommendations. We hope that you find our answers and explanations satisfactory.

Sincerely, on behalf of all authors,

                                                                                                Attila Borics

Reviewer 2 Report

The text discusses the investigation of the type I cannabinoid G protein-coupled receptor (CB1, GPCR) as a potential target for medication development. The activation mechanism of GPCRs is known to involve distinct functional states controlled by conformational switches. The text describes recent studies on the μ-opioid and β2-adrenergic receptors, which suggest a shift of macroscopic polarization in the transmembrane domain that is responsible for the activation of different functional states. The text also reports on a study of the CB1 receptor using all-atom molecular dynamics simulations, which confirmed the general features of the activation mechanism and discovered unique properties specific to this receptor.
The results are logically and clearly presented and discussed based on properly selected literature. I assess the whole work very well and conclude that the work meets all the requirements for publication in Internation Journal of Molecular Sciences.

Author Response

Answer to Reviewer 2.

English language and style

( ) English very difficult to understand/incomprehensible
( ) Extensive editing of English language and style required
( ) Moderate English changes required
( ) English language and style are fine/minor spell check required
(x) I don't feel qualified to judge about the English language and style

Yes

Can be improved

Must be improved

Not applicable

Does the introduction provide sufficient background and include all relevant references?

(x)

( )

( )

( )

Are all the cited references relevant to the research?

(x)

( )

( )

( )

Is the research design appropriate?

(x)

( )

( )

( )

Are the methods adequately described?

(x)

( )

( )

( )

Are the results clearly presented?

(x)

( )

( )

( )

Are the conclusions supported by the results?

(x)

( )

( )

( )

Comments and Suggestions for Authors

The text discusses the investigation of the type I cannabinoid G protein-coupled receptor (CB1, GPCR) as a potential target for medication development. The activation mechanism of GPCRs is known to involve distinct functional states controlled by conformational switches. The text describes recent studies on the μ-opioid and β2-adrenergic receptors, which suggest a shift of macroscopic polarization in the transmembrane domain that is responsible for the activation of different functional states. The text also reports on a study of the CB1 receptor using all-atom molecular dynamics simulations, which confirmed the general features of the activation mechanism and discovered unique properties specific to this receptor.
The results are logically and clearly presented and discussed based on properly selected literature. I assess the whole work very well and conclude that the work meets all the requirements for publication in Internation Journal of Molecular Sciences.

On behalf of all of the authors I sincerely thank you for your review and your supportive opinion. We are glad that our research and its presentation made such good impression. Since no concerns were raised, no further modifications of the manuscript have been made.

Sincerely,

                                                                                    Attila Borics